# Socially Responsible HRM, Employee Attitude, and Bank Reputation: The Rise of CSR in Bangladesh

Farid Ahammad Sobhani [1], Amlan Haque [2,*] and Shafiqur Rahman [3]

1   School of Business & Economics, United International University, Dhaka 1212, Bangladesh; drsobhani@uiu.ac.bd
2   School of Business & Law, Sydney Campus, CQUniversity, Sydney, NSW 2000, Australia
3   Higher Education, Kent Institute Australia, Sydney, NSW 2000, Australia; shafiqur.rahman@aabl.com.au
*   Correspondence: a.haque@cqu.edu.au; Tel.: +61-(2)-9324-5038

**Abstract:** Applying the organisational climate of corporate social responsibility (CSR) and human resource management (HRM) behavioural theories, this paper examines the associations among socially responsible HRM (SRHRM), organisational citizenship behaviour (OCB), turnover intention, and bank reputation. The proposed model, including seven hypotheses, was examined on a sample of 711 Bangladeshi bank employees. The results suggest that SRHRM has significant positive effects on both OCB and bank reputation, and a significant negative influence on turnover intention. OCB was positively significant for bank reputation but was estimated as insignificant for bank employees' turnover intention. Moreover, perceived bank reputation was significant and negative on Bangladeshi employees' turnover intention. Finally, a partial mediation effect was found for OCB on the direct relationship between SRHRM and bank reputation. This paper recommends that banks should aim at higher levels of OCB and reputation and lower turnover intentions should encourage SRHRM in their strategic approaches for HRM and CSR. The implications of the results of this study can help financial organisations to recognise the significance of SRHRM and its favourable effects on employee motivation and institutional reputation.

**Keywords:** socially responsible HRM; organisational citizenship behaviour; turnover intention; bank reputation; climate of corporate social responsibility; Bangladesh

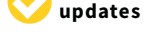



## 1. Introduction

The increased importance of socially responsible human resource management (SRHRM) leverages corporate social responsibility (popularly known as CSR) and employee motivation. The concept of SRHRM involved the aims of CSR into human resource management (HRM) and reflects stakeholders' expectations for financial, social, and environmental performance [1–3]. Currently, scholars recognise SRHRM as an inescapable priority for business leaders and recommended it as one of the essential factors in ensuring bank's competitive advantage and long-term sustainability [3–5]. Although SRHRM often may not have any short-term monetary benefits [6], it has proven to be a novel HRM approach for enhancing institutional reputation and workforce motivation outcomes in the mid to long term [2,5,7]. Scholars suggested that both CSR and general HRM can execute SRHRM to uphold employee motivation and organisational reputation [8–11]. Collier and Esteban [12] noted that employees "carry the main burden of responsibility for implementing ethical corporate behaviour in the daily working life of the company, (thus) the achievement of these outcomes depends largely on employee willingness to collaborate" (p. 20). Hence, the implication of SRHRM is critical to banks for their institutional reputation, and the promotion of CSR can improve employee work attitudes, such as citizenship behaviour and turnover intention.

Previous studies examined the influence of SRHRM on different employee attitudes and organisational outcomes, such as job satisfaction [4], employee wellbeing [13], job

commitment [3], organisational performance [6], profitability [5], and sustainability [14]. However, the exploration of the relationship between employee turnover intention and perceived organisational reputation is a gap in the HRM literature. For example, employees' turnover intention and citizenship behaviour within their organisation may be influenced by factors beyond individual attitudes, such as an organisation's CSR initiatives with SRHRM and perceived institutional reputation [3,6]. Hence, it is possible that, even when banks generally support HRM and try to adopt SRHRM, employees may or may not necessarily increase their organisational citizenship behaviour (OCB) or perceive higher bank reputation.

Employees show higher OCBs and lower turnover intentions in their work behaviours, even when they do not see enough CSR or HRM benefits in their organisational activities [5]. When employees are not representing their banks with a higher sense of OCB and indicating a willingness to switch their jobs, the question arises about the overall reputation and HRM effectiveness. Hence, banks need to gain positive OCBs from their employees and lessen their turnover intention to demonstrate a sound practice of SRHRM for higher reputation. Several scholars suggested that organisations can increase employee's motivational responses, for example, higher OCB and lower turnover intention, by formulating and implementing SRHRM practices [3,5,15]. SRHRM is a set of HRM practices adopted by organisations to affect employee attitudes and behaviours to facilitate CSR initiatives to benefit their employees, their stakeholders, and societies [10,11]. Scholars have recommended SRHRM practices for organisations to aid in their recruitment of socially responsible employees, their training for CSR practices, and to increase employees' social contributions to improve job performance for customers and stakeholders [5,16]. However, it remains unknown whether organisations can increase OCB sand decrease overall turnover through turnover intention by executing SRHRM practices. Several studies with SRHRM have been conducted in developed countries [6,17], but there remains a lack of empirical studies, particularly for a developing country like Bangladesh. Accordingly, this paper considers the following two research questions. First, is there a significant relationship between SRHRM and employee attitude (e.g., OCB, turnover intention, and bank reputation)? Second, is there a significant mediating relationship among SRHRM, OCB, turnover intention, and bank reputation? Therefore, this paper aims to examine the relationship between SRHRM and employee OCB and turnover intention for organisational reputation, and the underlying mechanisms among the Bangladeshi bank employees, to overcome this lack of literature. In other words, this paper will explore the influence of SRHRM on Bangladeshi bank employees, indicating their OCB, retention attitudes, and perceptions of their organisation's reputation.

## 2. The Banking Sector of Bangladesh, HRM Practices, and CSR Initiatives

As of 2020, there are 60 scheduled banks in Bangladesh, including three that are specialised, six that are state-owned commercial banks, nine foreign commercial banks, and 42 local private commercial banks [18]. Table 1 summarises the structure of theses scheduled banks. Besides, five non-scheduled banks such as Ansar VDP Unnayan Bank, Karmasangsthan Bank, Grameen Bank, Jubilee Bank, and Palli Shanchay Bank also operate in the banking sector [18]. As the central authority and regulatory body, "Bangladesh Bank" controls the banking sector's overall monetary and financial system. It was established in 1972 (with the corporate vide: P.O. No. 127 of 1972) with effect from 16th December 1971 [19].

**Table 1.** Banking system in Bangladesh.

| Type of Banks | No. of Banks | No. of Branches | | | Percentage of Total Assets (till Dec 2019) | Percentage of Total Deposit (till Dec 2019) |
|---|---|---|---|---|---|---|
| | | Urban | Rural | Total | | |
| State Owned Commercial | 6 | 1755 | 2019 | 3774 | 24.50 | 24.77 |
| Specialised | 3 | 278 | 1205 | 1484 | 2.19 | 2.49 |
| Local Private Commercial | 42 | 3352 | 1909 | 5261 | 67.79 | 68.22 |
| Foreign Commercial | 9 | 65 | 0 | 65 | 5.51 | 4.52 |
| Total | 60 | 5450 | 5133 | 10,583 | 100 | 100 |

Source: Bangladesh Economic Review [18].

Since independence was declared on 16 December 1971, the Government of Bangladesh has continuously supported the banking industry to boost its overall economic growth. As a result, to become a developed nation, the banking sector plays a leading role and contributes significantly to the job market to reduce the unemployment rate [20,21]. At present, attracting job seekers to banking careers is becoming more of a challenge. HRM plays a critical role in fulfilling future banking job prospects, including employees' motivation and their stakeholders' expectations. For example, banks are continuously competing within the industry to become more reputed for their sustainability and profitability [22]. Scholars recommend that SRHRM should focus more on CSR for banks to improve their reputation and employee performance and enhance socially responsible behaviours for long-term sustainability and social welfare [20]. Therefore, HRM that focuses on CSR initiatives has become a priority strategy for Bangladeshi banks [21]. Consequently, Bangladeshi banks are expending more on their CSR-related initiatives. Table 2 summarises the CSR expenditures highlighted from the last five years of expenses of banks in Bangladesh (2016 to 2020).

**Table 2.** Banks' CSR spending in Bangladesh.

| | CSR Spending | | | | Figures in Million USD | | | |
|---|---|---|---|---|---|---|---|---|
| Year | Education | Health | Disaster Relief | Environment | Culture | Others | Total | |
| 2020 | 22.62 | 18.25 | 32.80 | 5.44 | 13.03 | 23.58 | 115.72 | |
| 2019 | 7.88 | 3.33 | 16.74 | 0.47 | 0.17 | 4.49 | 33.08 | |
| 2018 | 43.53 | 5.54 | 60.27 | 0.98 | 3.68 | 16.71 | 130.71 | |
| 2017 | 23.95 | 7.08 | 22.96 | 1.24 | 3.58 | 8.87 | 67.68 | |
| 2016 | 20.14 | 9.45 | 22.37 | 1.34 | 3.83 | 12.17 | 69.30 | |
| Total | 118.12 | 43.65 | 155.14 | 9.47 | 24.29 | 65.82 | 416.49 | |

Source: Bangladesh Bank [19].

## 3. Theoretical Background

This paper aims to explore SRHRM, considering bank reputation as the extent to which employees perceive their bank's overall reputation and are willing to show more OCBs and less turnover intention. SRHRM for CSR initiatives mostly addresses issues linked to economic benefits for society through tackling poverty reduction [6], ecological sustainability [23], global climate change [24], and social engagement, such as through the implementation of educational scholarships and entrepreneurial or start-up loans with minimum interest. However, a bank's efforts towards SRHRM do not always align with employees' motivations for job effort or organisational interests [25]. Scholars suggest that

the effects of SRHRM on employee attitudes and behaviours may vary with the attributes of CSR [3,6,26]. Nishii et al. [27] noted that "for HR practices to exert their desired effect on employee attitudes and behaviours, they first have to be perceived and interpreted subjectively by employees in ways that will engender such attitudinal and behavioural reactions" (p. 504). Accordingly, this paper suggests that SRHRM will positively influence employee OCB and negatively affect the turnover intention to increase perceptions of bank reputation.

Following the literature on organisational CSR climates [28], this paper argues that SRHRM is associated with employee's OCB and turnover intention for corporate reputation through social and psychological processes. Reichers and Schneider [28] described the theory of organisational CSR climates as being employees' collective perceptions of how HRM helps organisations to deliver various stakeholders' interests. Moreover, this paper relies on the stakeholder theory [29] to explore the studied relationships among SRHRM, OCB, employee turnover, and bank reputation. According to the stakeholder theory [29], this paper claims that banks serve numerous stakeholders with various interests and conflicting demands for CSR activities. Hence, the reactions of stakeholders such as employees, shareholders, and communities to a bank's SRHRM practices are likely to be centred around how their interests are met and the perceptions of the reputation of the bank. Accordingly, this paper argues that the influence of SRHRM on bank reputation depends on the outcomes of employees' OCB and turnover intention and how the organisations treat their stakeholders in terms of CSR practices.

This paper contributes to the literature of HRM and organisational studies in several ways. First, as a newly developed concept, SRHRM has received significant attention from both HRM scholars and practitioners. For example, researchers have examined the influence of SRHRM both on the employee [2,13] and organisational outcomes [5,14]. More-over, SRHRM evolved from economics and includes strategic management and corporate governance [10,11]. However, there is limited empirical evidence in the literature about the impact of SRHRM in understanding employee attitudes such as OCB and turnover intention [7,30]. Hence, the employee-level outcomes for SRHRM remain relatively unexplored. By examining the impact of SRHRM on employee's OCB and turnover intention and bank reputation, this paper considers the effects of both employee-level perception and organisational context.

Second, this paper extends CSR's call through SRHRM into the literature of HRM [30]. The concept of SRHRM has evolved in the last two decades and received limited attention with regards to empirical studies [1,16]. At the early stage of the idea, Shen and Zhu [1] justified the significance of SRHRM and examined its association with employee organisational commitment. Similarly, Shen and Benson [16] explored the effect of SRHRM on work performance and employee support behaviour. However, several scholars criticised SRHRM for not having enough research regarding employee outcomes for organisational performance and sustainability [14,26,30,31]. For example, Nishii et al. [27] suggested examining SRHRM to improve employee attitudes and behaviours for societal contribution, as employees' consequences of SRHRM have not been studied enough for reputational purposes previously. This paper develops the conceptual framework (see Figure 1) based on Reichers and Schneider [28] and recommends that the organisational climate in banks should include psychological processes for HRM policies and practices to promote SRHRM [32,33]. This conceptual model also tests a moderation/mediation of OCB, which has not been previously studied in SRHRM research. Hence, this study extends the current knowledge of SRHRM and its employee consequences with the justifications of the social and psychological processes of Reichers and Schneider [28].

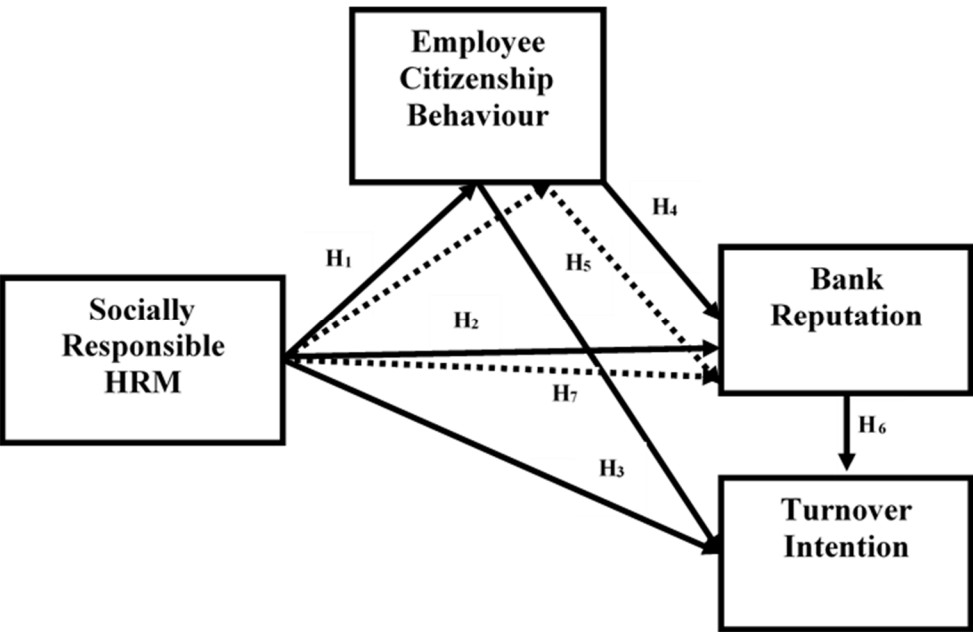

**Figure 1.** Hypothesised model proposing direct (solid lines) and mediational (dotted lines) relationships.

Third, employees' responses to their organisation's social responsibility have been less observed and require further study [9]. This paper helps to overcome this limitation in the following ways. First, examining the impact of SRHRM on an employee level (e.g., OCB and turnover intention) and then at an institutional level (e.g., reputation). Second, exploring the effects of OCB and turnover intention on a bank's reputation, and third, recognising potential conflicting interests of employee's OCB and turnover intention to justify the current practice of HRM to establish SRHRM for higher reputation. Hence, this paper aims to understand the behavioural and psychological perspectives of employees to understand why and when SRHRM influences OCB and turnover intention for their banks' social responsibility targeted future reputation.

Finally, this paper examines the proposed model in the context of Bangladesh, which has successfully achieved the mark of a developing nation because of its fast economic growth. The banking sector of Bangladesh has played a significant role in this journey and the maintenance of its successful performance as one of the countries' leading sectors is of paramount importance. This sector is managed by one central bank (Bangladesh Bank). It includes three specialised developments, five non-scheduled, six state-owned, forty-one private, and nine foreign commercial banks [34]. In recent times, the Government of Bangladesh has emphasised social engagement through CSR to enable the transition to a digital Bangladesh for higher social engagement and contribution. As a result, Bangladeshi banks are more inclined to implement SRHRM practices [35,36]. Therefore, Bangladesh is a suitable country in which to conduct this research on SRHRM. Accordingly, the study findings will have practical implications for all the Bangladeshi banks irrespective of their above categorial differences.

## 4. Hypotheses Development

### 4.1. Socially Responsible HRM (SRHRM), OCB, and Turnover Intention

Employees' behavioural perspective on overall HRM practices impacts their work attitudes and interprets the organisational performance of SRHRM [5,32,33]. Banks adopt various approach to HRM practices for specific financial goals and purposes. According to the attribution theory [37], employees hold the inclination to analyse the causes and effects of the behaviour of their other colleagues and managers. For example, an employee's perception of another individuals' behaviour and actions in the same organisation influences their future behaviour and perception. Researchers argued that employees tend to

personalise their organisations' HRM practices and adjust their attitudinal (e.g., OCB) and behavioural (e.g., turnover intention) responses [27,31]. Hence, following the attribution theory [37] and the perceptions regarding how bank employees are treated by the banks largely, this paper argues that a bank's policies and practices of HRM significantly affect employees' work attitudes and behaviours [27] and determine OCB and turnover intention.

The notion of OCB has been identified as a critical construct in psychology and management literature and is considered an essential factor contributing to an organisation's survival [38]. Researchers discovered that employees who keep more OCB are expected to have higher levels of job motivations that lead to increased productivity, less staff turnover, and higher profitability for organisations [26,39,40]. On the other hand, turnover intention refers to employees' thoughts and awareness about leaving the job, leading to actual voluntary turnover [41,42]. Due to the prospect of better careers and a higher number of banks in Bangladesh, bank employees enjoy better job opportunities in the industry which may provoke higher turnover intention. As a unique concept, SRHRM includes CSR's value to expedite CSR policies such as recruiting employees who have positive attitudes toward CSR and training to meet the bank's stakeholders' expectations. Ellis [43] indicated that a lack of focus on CSR and its implication could negatively affect the motivation and organisational performance of employees. Hence, SRHRM implementing CSR training can enhance employees' motivation and awareness of CSR and improve their social participation, such as voluntary work, donations, and community involvement [2,15]. A good number of studies have shown the significant relationships between the practice of HRM with OCB [7,16] and turnover intention [17–19]; however, there is a lack of empirical evidence for SRHRM for Bangladeshi bank employees. These lead to the following two hypotheses.

**Hypothesis 1 (H$_1$).** *SRHRM practices will be positively related to employee OCB.*

**Hypothesis 2 (H$_2$).** *SRHRM practices will be negatively related to employee turnover intention.*

### 4.2. Socially Responsible HRM (SRHRM) and Bank Reputation

Bank (i.e., corporate or organisational) reputation refers to a collective set of observations and beliefs by stakeholders that work as a long-term asset for organisational competitiveness [20]. It has been considered an intangible corporate asset or investment that secures a competitive position and assures long-term sustainability [21,22]. Scholars have suggested that corporate reputation can influence or uphold consumer behaviour [24], organisational performance [44], profitability [22], and sustainability [23]. It allows organisations to demand higher premiums for their offered products or services [20,24], helps them to attract and retain employees, and increases OCB [25,26]. Scholars have also recommended strategic HRM to enhance organisational reputation by engaging the senior executives and HR managers [1,27]. According to Friedman [45], "HR activities can positively impact organisational reputation, but how HRM specifically influences particular dimensions of reputation needs to be explored" (pp. 229–230). Moreover, when employees perceive higher reputation through organisational performance, then understandably, the bank will benefit from higher employee retention and sustainable HRM [10,23]. Consequently, previous studies examined the relationship between HRM and organisational reputation [16,45]; however, there is a lack of empirical evidence for SRHRM. Hence, in line with the above studies, this paper extends the literature of SRHRM and offers the following hypothesis:

**Hypothesis 3 (H$_3$).** *SRHRM practices will be positively related to employee's perceived bank reputation.*

### 4.3. Associations among OCB, Turnover Intention, and Bank Reputation

Bateman and Organ [38] reported on OCB, and Organ [46] defined it as "individual behaviour that is discretionary, not directly or explicitly recognised by the formal reward system, and that in the aggregate promotes the effective functioning of the organisation"

(p. 4). It goes beyond employees' formal obligations and roles to support their businesses and may offer a valuable contribution to higher employee motivation and organisational attachment. Researchers have examined critical role of OCB in employee performance, job satisfaction, and turnover intention [47,48]. Studies have indicated a negative relationship between OCB and the turnover intention of employees [48–50]. This is because, employees displaying more OCBs showed more effort to fulfil their job requirements, a higher tolerance for t inconvenience, and a sense of organisational belonging that reduces turnover intention [48,50,51]. According to the cognitive consistency theory [52], OCB helps to engage employees in a state of synchrony to response against voluntary turnover by actively avoiding conflict between their beliefs and behaviours. Hence, employees with higher OCB demonstrate their intention or willingness to stay longer within their organisations [49]. Accordingly, this paper suggests the following hypothesis:

**Hypothesis 4 (H$_4$).** *OCB will be negatively related to turnover intention.*

Employees who have more OCBs are more committed to accepting organisational challenges and perceive a higher reputation of their organisations [53,54]. Following Hypothesis 1 (H$_1$), this paper suggests that employees who have more OCBs not only show less turnover intention but also contribute to developing a reputation based on such motivations such as job satisfaction, tolerance for uncertainty, and job performance [54,55]. According to Hall et al. [53], OCB increases employees' perceptions regarding organisational reputation via the sense of belief, trust, and links to job performance through social identification. Several scholars reported that employees' OCB helps to increase their impression management and attempts to increase their sense of organisational reputation [54,55]; and therefore, this paper considers the following hypothesis:

**Hypothesis 5 (H$_5$).** *OCB will be positively related to employee's perceived bank reputation.*

*4.4. Relationship between Bank Reputation and Turnover Intention*

The notion of turnover intention has commonly been considered an employee's willingness to withdraw from the organisation and is perceived as the most vital cognitive precursor of actual turnover [41,56]. However, it does not necessarily translate into actual employee turnover but is recognised as a significant predictor of actual turnover [41]. Scholars described turnover intention as employee's subjective estimation that includes three states of mind; thoughts of resigning from the job, making the choice to search for a new job, and then deciding to quit [57]. This paper argues that working for an organisation with a higher reputation may increase employees' psychological and motivational engagement with their organisation, consequently reducing their turnover intention. Numerous studies explored that working for an organisation with a higher reputation enhances employees' job satisfaction and social identification and reduces turnover intention [58,59]. According to the social identity theory [60], employees' perception of their organisation's reputation increases their sense of belonging both within their job and the organisation, reducing turnover intention [41,60]. Accordingly, this paper includes the following hypothesis:

**Hypothesis 6 (H$_6$).** *Employee's perception of bank reputation will be negatively related to turnover intention.*

Following the Hypotheses 1, 2, and 4 (H$_1$, H$_2$, and H$_4$), this paper argues that when employees display more OCB, they are likely to perceive a higher level of bank reputation because of their mediational impact on the direct relationship between SRHRM and bank reputation. According to the job-demand resources (JDR) theory [61], the effect of SRHRM can enhance employee motivation and attitude, enabling them to be more engaged with their work to increase bank performance. As a result, the increased overall bank performance can lead to higher profits and employees may perceive a higher bank reputation. Hence, this paper argues for a mediation effect of OCB on the direct relationship between

SRHRM and bank reputation. However, to fulfil the condition of a mediational impact, two states need to be met as a rule of thumb. First, the significant associations among SRHRM, OCB, and bank reputation should be included. Second, the inclusion of OCB in the direct relationship between SRHRM and bank reputation should increase or decrease their previous indicative value. Accordingly, this paper proposes the following hypothesis:

**Hypothesis 7 (H7).** *OCB mediates the direct relationship between SRHRM and bank reputation.*

In relation to the discussion above, Figure 1 depicts both the direct and mediational relationships and their associated hypotheses.

## 5. Methods

### 5.1. Sample and Procedures for Data Collection

This study administered a two-step online survey with structured questionnaires, including measures for SRHRM, OCB, turnover intention, and bank reputation. At the first step, a pilot test with 75 responses was examined to observe the designed survey's errors or issues. No issues were identified except some minor changes to the English and Bengali translated versions of the wording. For example, the word "organisation" was replaced with "bank", and "remuneration" with "salary and benefits". At the second step, this study conducted the data collection among Bangladeshi bank employees above 18 years of age and continuing their job as full-time employees. A total of 818 responses were collected from 57 banks, including 48 private and nine public. The questionnaires, including the English with Bengali translation, were distributed online and through internal bank communications from within Dhaka city. The final sample size (N = 711) for this study deemed sufficient by a power analysis (effect size 0.15 and error probability 0.05).

The data collection was accomplished, ensuring all the required privacy and confidentiality, and participants had the choice to withdraw their responses at any time. Estimating a response rate of 23.37% in the demographic profile, 607 (85.4%) were male, and 104 (14.6%) were female. In the age category, the largest group, 414 (58.2%), were aged between 26 and 35 years; 194 (27.3%) were aged between 36 and 45; and 14 (2%) of the respondents were between 56 and 65. For the duration of service, the largest group, 209 (29.4%), had 4 to 7 years of job experience, followed by 176 (24.8%) with 1 to 3 years and 148 (20.8%) with 8 to 11 years of job experience. Most of the bank employees, 608 (85.5%), in this study had master's or MBA degrees, including 100 (14.1%) for BBA. Only 3 (0.4%) earned the doctoral level of education. Finally, 153 (21.5%) were earning between BDT 200,000 to 400,000 annually (after tax), which is approximately USD 2380–4761 (considering 1 USD = 84 BDT on 8th February 2021), and 121 (17%) were earning up to BDT 800,000 (i.e., USD 9523). However, 121 (17%) of the respondents preferred not to share their annual income.

### 5.2. Measures

This study applied a five-point Likert-type scale (e.g., 1 = strongly disagree to 5 = strongly agree) for all study variables such as SRHRM [1], OCB [62], turnover intention [63], and bank reputation [64]. The finalised survey questionnaire was drafted in English and translated into Bengali to understand the survey questions better.

#### 5.2.1. SRHRM Practices

The 13-item scale for SRHRM practices was collected from Shen and Zhu [1]. It included three subscales such as legal compliance HRM (LC-HRM; six items), employee-oriented HRM (EO-HRM; four items), and general CSR facilitation HRM (GF-HRM; three items). A sample item from LC-HRM was "my bank ensures equal opportunity in HRM". For EO-HRM, "my bank provides adequate training and development opportunities to employees". Lastly, for GF-HRM, "my bank rewards employees who contribute to charity, communities, and other CSR activities". The estimated Cronbach's alpha of SRHRM in this study was 0.88.

#### 5.2.2. OCB

An eight-item scale for OCB was used, taken from Lee and Allen [62]. A sample item from the scale was "I attend functions that are not compulsory, but that help the organisational image." The estimated Cronbach's alpha for the scale in this study was 0.79.

#### 5.2.3. Turnover Intention

Turnover intention for the Bangladeshi bank employees was measured with a four-item scale included from Kelloway et al. [63]. A sample item was "I am thinking about leaving this bank". The Cronbach's alpha of the scale in this study was 0.93.

#### 5.2.4. Bank Reputation

The scale for bank reputation for this study was collected from Lin et al. [64]. One of the three items for this scale was "my bank is a reputable organisation to work for". The Cronbach's alpha of this scale in this study was 0.84.

### 5.3. Data-Analysis Procedures

A set of statistical interventions such as the means, standard deviations, correlation matrix, and reliability analyses of the adopted scales were administered to test the hypothesised model (Figure 1). Following Anderson and Gerbing [65], this paper conducted a two-step structural equation modelling (SEM) technique for the data analysis and hypotheses testing. First, the measurement model examined the scales' acceptability for their internal errors (e.g., composite scale reliabilities and discriminant validities). The second step of SEM evaluates the overall associations among the variables (e.g., SRHRM, OCB, turnover intention, and reputation) by outlining how they fit into the model using path analysis [66]. AMOS (IBM SPSS Statistics 26) was used to examine the over-fit indices (e.g., the norm fit index: root-mean-square error of approximation and $\chi^2$) and the values of structural coefficients. Raykov and Marcoulides [67] recommended SEM for its reliability and consistent path analysis ability. This paper examined the mediation impact of OCB on the relationship between SRHRM and turnover intention with further investigation.

### 6. Results and Hypothesis Testing

#### 6.1. Descriptive Statistics and Correlation Test

Table 3 summarizes the descriptive statistics including correlations and Cronbach's alpha values for all the studied variables. SRHRM showed positive and significant correlations with OCB (0.479 **, $p < 0.01$) and bank reputation (0.575 **, $p < 0.01$), and a negative relationship with turnover intention (−312 **, $p < 0.01$). OCB showed strong and significant positive correlations with bank reputation (0.531 **, $p < 0.01$), and a negative influence on turnover intention (−0.215, $p < 0.01$). Lastly, employee perception of bank reputation indicated a significant negative correlation with turnover intention (−338 **, $p < 0.01$). Principally, this paper found higher and significant correlations among all the studied variables, showing that the initial indicators support the above Hypotheses 1 to 7.

**Table 3.** Descriptive statistics and Pearson correlation coefficients.

|  | M | SD | Skewness | SRHRM | OCB | CORP-REPU | TURN-INT |
|---|---|---|---|---|---|---|---|
| SRHRM ($\alpha$ = 0.88) | 3.94 | 9.25 | −0.94 | 1 |  |  |  |
| OCB ($\alpha$ = 0.79) | 4.65 | 3.62 | −2.52 | 0.479 ** | 1 |  |  |
| Bank reputation ($\alpha$ = 0.84) | 4.59 | 1.93 | −2.15 | 0.575 ** | 0.531 ** | 1 |  |
| Turnover intention ($\alpha$ = 0.93) | 2.51 | 4.87 | 0.30 | −0.312 ** | −0.215 ** | −338 ** | 1 |

** $p < 0.01$ level.

### 6.2. First Step: Measurement Model

First, this paper estimated SRHRM with a confirmatory factor analysis (CFA). Accordingly, the scale showed appropriate fit with the proposed measurement model ($\chi^2$ = 312.75, $\chi^2$/df = 5.30, $p$ = 0.000, GFI = 0.94, AGFI = 0.90, CFI = 0.93, TLI = 0.91, NFI = 0.91, RMSEA = 0.078, and SRMR = 0.0439). Results from the CFA indicated significant loadings for SRHRM (from 0.60 to 0.90). Second, OCB had eight items and showed a satisfactory fit ($\chi^2$ = 119.38, $\chi^2$/df5 = 6.28, $p$ = 0.000, GFI = 0.96, AGFI = 0.92, CFI = 0.95, TLI = 0.93, NFI = 0.95, RMSEA = 0.086, and SRMR = 0.0423). Third, turnover intention included four items (loading from 0.81 to 0.94) and indicated a good fit ($\chi^2$ = 19.27, $\chi^2$/df5 = 9.64, $p$ = 0.000, GFI = 0.98, AGFI = 0.94, CFI = 0.99, TLI = 0.97, NFI = 0.99, RMSEA = 0.110, and SRMR = 0.0125). Lastly, the scale for bank reputation had three items and the CFA for bank reputation estimated an over-fit due to its three items, with GFI = 1, CFI = 1, NFI = 1 and RMSEA = 0.651 and SRMR = 0.000. A composite reliability [68] was calculated from the structural model (Figure 2) with each of the loadings' (e.g., 0.67, 0.87, and 0.84) standardised values for bank reputation, and a reliability score of 0.84 was found for the scale. Scholars suggested that an RMSEA ranging within 0.05 to 0.10 and others over 0.80 can be considered a good fit [66,69].

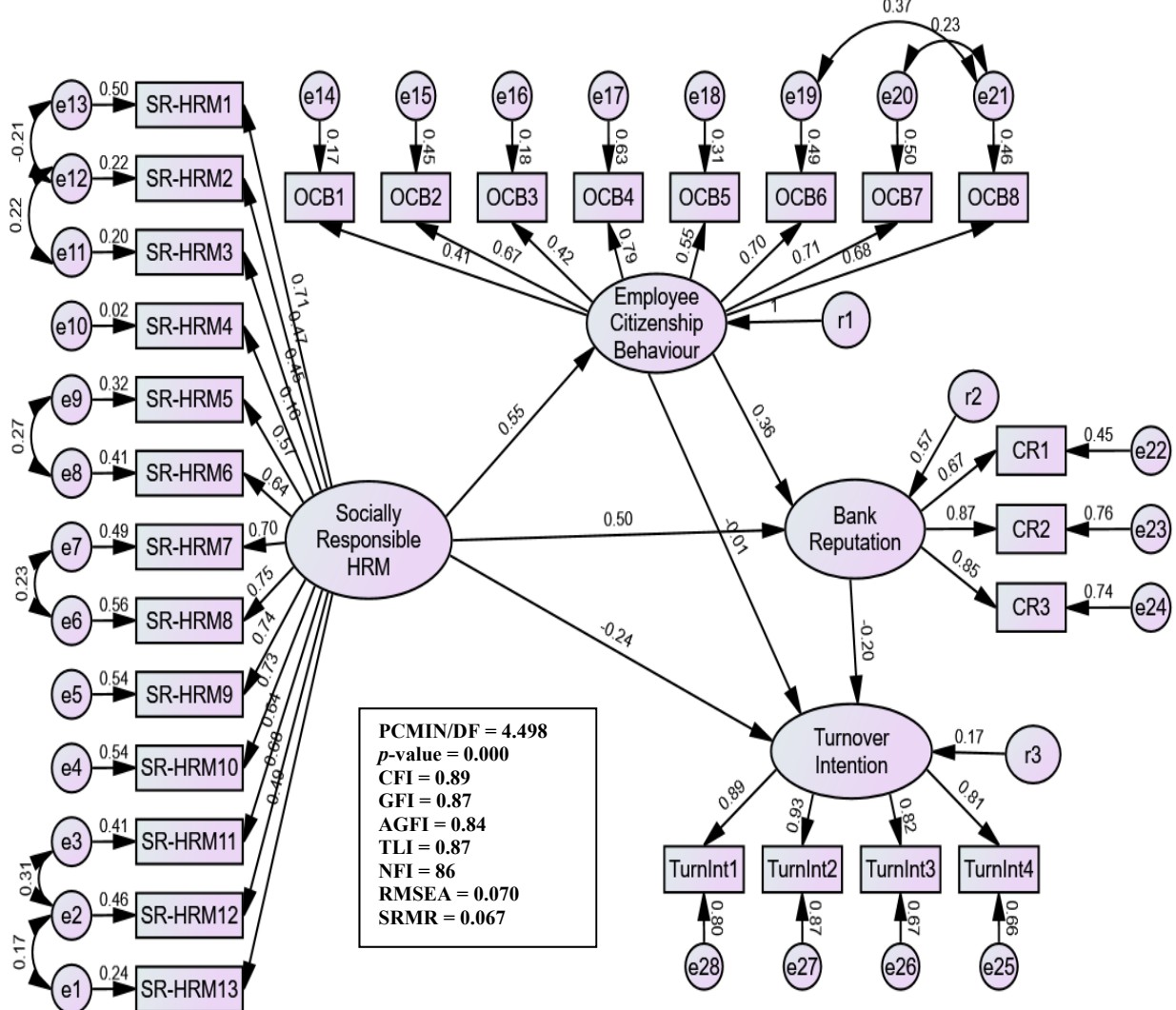

**Figure 2.** Indirect relationships among SRHRM, OCB, bank reputation, and turnover intention. Note: $n$ = 200. Bootstrap (i.e., resample) = 5000, percentile and bias-corrected confidence intervals are on 99%.

*6.3. Second Step: Structural Model*

This paper completed all the necessary calculations for the second step of Anderson and Gerbing's [65] SEM technique. Figure 2 reflects the confirmatory factor analysis (CFA) as proposed in the hypothesised model (Figure 1) with the necessary regression weights. Results from the CFA were found to have sufficient evidence ($\chi^2$ = 1544.214, $\chi^2/df5$ = 4.45, $p$ = 0.000, GFI = 0.86, AGFI = 0.83, CFI = 0.89, TLI = 0.87, NFI = 0.89, RMSEA = 0.070, and SRMR = 0.067) to claim a good fit for the data with the proposed model (Figure 1). Therefore, this paper met both the steps of Anderson and Gerbing [65] to suitably fit and justify the relationships among SRHRM, OCB, bank reputation, and turnover intention among the Bangladeshi bank employees.

*6.4. Hypothesis Testing*

SEM has been recognised as a useful statistical tool for testing hypothesised models [41,66]. This paper applied Lindell and Whitney's marker variable technique to scrutinise and eliminate the common method bias (CMB). It also examined the multicollinearity, variance, and normality of the study data and their tolerance values ($1-R^2$) to overcome the limitations of the CMB and inflation factors, finding their outcomes satisfactory. For example, Harman's single factor test for the total variance estimated 29.17% to be the standard limit of the cut-off point noted up to 50%. Similarly, this study's skewness was between $-0.94$ to 0.30 and was found to have a normal distribution over the overall dataset. Hence, this study claims no consequence of CMB for the examined hypotheses.

As shown in Figure 2, Hypotheses 1, 2, and 3 ($H_1$, $H_2$, and $H_3$) were formulated to exam the nature of SRHRM on OCB, bank reputation, and turnover intention, respectively. Here, the beta value (β) indicated a significant and positive association between SRHRM and OCB (β = 0.348; $p < 0.001$) and bank reputation (β = 0.351; $p < 0.001$). However, a significant and negative relationship was estimated between SRHRM and turnover intention (β = $-0.404$; $p < 0.001$). Accordingly, $H_1$, $H_2$, and $H_3$ were accepted in this study. This paper supported the associations among OCB, bank reputation, and turnover intention with the Hypotheses 4 and 5 ($H_4$ and $H_5$). Here, the relationship between OCB and bank reputation was significantly positive (β = 0.358; $p < 0.001$), and accepted $H_4$. However, the positive relationship between OCB and turnover intention was rejected because of their insignificant $p$-value (β = $-0.038$; $p < 0.799$). The impact of bank reputation on turnover intention was described with Hypothesis 6 ($H_6$) and a significant negative relationship was found (β = $-0.521$; $p < 0.004$), therefore, this paper accepted $H_5$ accordingly. Table 4 represents the findings linked to the hypothesised model of this study (see Figure 1).

**Table 4.** Summary of the results.

| Hypothesis | Relationship | Beta Value (β) with Significance (*p*-Value) | Results of Analysis |
|---|---|---|---|
| $H_1$ | SRHRM on OCB | (β = 0.348; $p < 0.001$) | Accepted |
| $H_2$ | SRHRM on bank reputation | (β = 0.351; $p < 0.001$) | Accepted |
| $H_3$ | SRHRM on turnover intention | (β = $-0.404$; $p < 0.001$) | Accepted |
| $H_4$ | OCB on bank reputation | (β = 0.358; $p < 0.001$) | Accepted |
| $H_5$ | OCB on turnover intention | (β = $-0.038$; $p < 0.799$) | Rejected |
| $H_6$ | Bank reputation on turnover intention | (β = $-0.521$; $p < 0.004$) | Accepted |
| $H_7$ | Mediational influence of OCB on the relationships between SRHRM and bank reputation | Direct influence of SRHRM on bank reputation increases from (β = 0.197; $p < 0.000$) to (β = 0.497; $p < 0.000$) (i.e., +0.30) | Partially mediated |

According to Hypothesis 7 ($H_7$), OCB partially mediates the relationship between SRHRM and bank reputation. This mediational impact demonstrates the significant role of OCB that determines the usefulness of SRHRM to promote a higher level of bank reputation. This outcome extends the previous claims of similar studies [60,61] and suggests that OCB

plays a decisive role in the application of SRHRM and employee perception of bank reputation. Table 5 summarises the mediational results of this paper.

**Table 5.** Mediational analysis.

| Mediational Influence of OCB | S. E. | C.R | Standardised Effect before Mediation (*p*-Value) | Standardised Effect after Mediation (*p*-Value) | Increased Effects | Mediational Effect |
|---|---|---|---|---|---|---|
| SRHRM → Bank Reputation | 0.036 | 8.756 | 0.197 (0.000) | 0.497 (0.000) | +0.30 | Partial |
| Fit Indices → $\chi^2$ = 1268.975, $\chi^2$/df5 = 5.27, *p* = 0.000, GFI = 0.84, AGFI = 0.83, CFI = 0.87, TLI = 0.85, NFI = 0.84, RMSEA = 0.078, and SRMR = 0.0715 | | | | | | |

### 6.5. Control Variables

This paper analysed age, gender, and duration of job experience (tenure) as control variables and examined their influences on the study results (Rodrigo and Arenas, 2008) [70]. However, none of these variables confounded the associations for the proposed model (Figure 1) except age on turnover intention ($p < 0.001$) and tenure on OCB ($p < 0.009$). According to Ferres et al. [71], both the values are justified because may older employees show less intention to leave their jobs because of the risk of fit between their needs and new jobs. Consequently, they will have more OCB for their banks. These findings are also supported by Krumm et al. [72]. Table 6 summarises the results for these control variables.

**Table 6.** Estimation of the control variables.

| Dimensions | | | Estimate | S. E. | C. R. | P |
|---|---|---|---|---|---|---|
| OCB | <- - - | AGE | 0.041 | 0.229 | 0.177 | 0.859 |
| OCB | <- - - | GENDER | −0.314 | 0.334 | −0.939 | 0.348 |
| OCB | <- - - | TENURE | 0.371 | 0.142 | 2.612 | 0.009 |
| Corporate Reputation | <- - - | AGE | 0.203 | 0.108 | 1.892 | 0.058 |
| Corporate Reputation | <- - - | GENDER | 0.183 | 0.157 | 1.166 | 0.243 |
| Corporate Reputation | <- - - | TENURE | −0.054 | 0.067 | −0.804 | 0.421 |
| Turnover Intention | <- - - | AGE | −1.324 | 0.319 | −4.158 | 0.001 |
| Turnover Intention | <- - - | GENDER | −0.001 | 0.464 | −0.001 | 0.999 |
| Turnover Intention | <- - - | TENURE | −0.187 | 0.197 | −0.949 | 0.343 |

## 7. Discussion

### 7.1. Findings of the Study

This paper explored the relationships among SRHRM, OCB, bank reputation, and turnover intention among Bangladeshi bank employees. The overall results from SEM indicate the significance of SRHRM and contribute to the literature of HRM in several ways. First, from Hypotheses 1 and 2 (H$_1$ and H$_2$), SRHRM was positively and significantly linked to OCB and bank reputation. These associations describe the value of SRHRM practices such as recruitment and selection, retention, training, and development in integrating CSR practices into Bangladeshi banks. According to the organisational CSR climate [28], this paper suggests that SRHRM can help organisations develop more meaningful relationships with their stakeholders by incorporating CSR values into their bank performance to increase employees' OCB and reputation. These findings follow Zhao et al. [11] and Shen and Zhang [2] to signify the importance of the SRHRM and its impact on OCB and employees' perception of bank reputation.

Hypothesis 3 (H$_3$) found that SHRM was significantly negative relationship to employee turnover intention. This outcome forecasts the capacity of SRHRM to influence future turnover [41] by upholding the practice of CSR-based HRM to reduce employee's intention to leave the organisation. This result supports and expands upon the theory of organisational CSR climates [28] by indicating that SRHRM influences employees' at-

tachment to their organisations and may reduce turnover for Bangladeshi banks. Similar studies such as [41,73] also recommend SRHRM for lower employee turnover.

Hypotheses 4 ($H_4$) showed a significant positive impact of OCB on bank reputation. According to Reichers and Schneider's [28] concept of organisational CSR climates, OCB has the positive effect of motivating employees and increasing perceptions of their bank's reputation, increasing their emotional attachment to the organisation to reduce turnover intention. However, Hypotheses 5 ($H_5$) showed a negative but insignificant ($\beta = -0.038$; $p < 0.799$) impact of OCB on turnover intention in this study. Hence, the rejected $H_5$ indicates that Bangladeshi bank employees ignore OCB to reduce turnover intention. Accordingly, this paper suggests that compared to SRHRM or bank reputation (e.g., $H_6$), OCB is less useful to reduce the turnover intention of employees of Bangladeshi banks These findings align with Lam et al. [51].

As suggested in Hypotheses 6 ($H_6$), an increase in employees' perception of their bank's reputation indicated a significant negative influence on turnover intention. This finding suggests that if the employees perceive a higher reputation for their banks, then they show less intention to leave their jobs. According to Reichers and Schneider [28], CSR's climate helps to satisfy the stakeholders and motivates employees to be more engaged with their work and the organisation. This finding reflects the suggestions of [59].

Finally, Hypotheses 7 ($H_7$) recommends that OCB partially mediates the relationship between SRHRM and bank reputation. This mediational result suggests the significant role of OCB that determines and upholds the impact of SRHRM to increase bank reputation. This result extends the previous literature of HRM studies [19,41,51,74–76] and suggests that the OCB of bank employees can be vital for Bangladeshi banks to increase the overall reputation of the financial sector.

### 7.2. Contributions to Management Practice

The call for SRHRM for stakeholders has significant implications for banks striving to increase their CSR initiatives. Previous HRM studies (e.g., [2,11]) provide a definite recommendation for how to undertake SRHRM to benefit organisational outcomes and outcomes related to employee motivation. Accordingly, this paper offers the following implications, particularly for financial institutions. First, a bank should adopt SRHRM to increase employee participation and motivation for CSR and to meet their stakeholders' higher demands. For example, more focus on OCB or turnover intention can be attained by hiring job candidates with better knowledge of CSR and the banks' values. More training related to CSR and employees' social performance can be practiced in performance appraisals, rewards, or promotion decisions. These initiatives are likely to enhance the practice demand of SRHRM and change employees' attitude toward CSR, increasing the bank reputation.

Second, banks should attempt to uphold OCBs and reduce employees' turnover intention to increase internal job performance and institutional reputation. This paper suggests that if the employees display more OCBs and have less intention to leave, banks are likely to establish SRHRM and succeed in making a better return of investment from their overall CSR initiatives. Moreover, banks should consider the capacity building of managers to aid them in motivating and engaging employees for higher bank reputation [76,77]. For example, banks may organize specific training and development sessions to help managers learn and apply SRHRM.

Lastly, banks have multiple stakeholders with conflicting interests and demands for CSR. This paper suggests that stakeholders' response is significantly dependent on how a bank's internal HRM shows the value of CSR and fulfils the needs of their societies. Hence, it will be critical for banks to achieve SRHRM to increase employees' higher level of OCB and decrease turnover intention to meet the future CSR demands and expectations for their employees and stakeholders, resulting in reputation gains.

### 7.3. Research Limitations and Future Research Opportunities

The demand and recognition for SRHRM and CSR practices vary among countries because of their organisational and governmental regulations [1,2,8]. For example, issues regarding employee wellbeing and environmental concerns in south Asian countries may not be treated as they would be in European or other Western countries. Accordingly, some limitations of this paper can be highlighted. First, this study was conducted in the Bangladeshi context, and the findings may not be equally applicable to other countries because of various socio-economic and financial regulations. Moreover, this paper did not consider the influence of cultural norms, such as collectivist or individualistic culture in Bangladesh [22,74]. Second, this paper considered the employees' perspectives and did not take into account managerial or stakeholder views. Hence, future researchers may explore the hypothesised model (Figure 1) from an executive or societal perspective. Third, this paper included OCB as a mediator; however, other factors, such as employee commitment or turnover intention [41,42] can also be considered mediators for future studies. Lastly, this study has investigated the relationships among SRHRM, OCB, turnover intention, and bank reputation. Future researchers may also explore the impact of SRHRM on other motivational and psychological factors for employees, such as employee wellbeing [13] or leadership responses [41].

### 8. Conclusions

This paper explored the impact of SRHRM among Bangladeshi bank employees to advance the current knowledge of HRM and contribute to the literature on organisational studies in general. The study results reveal that (a) SRHRM significantly influences employees' OCB, turnover intention, and perception of bank reputation. However, (b) OCB's influence on employees' turnover intention was found to be insignificant but positive on their perception of bank reputation. Moreover, (c) the direct relationship between SRHRM and bank reputation found to be partially mediated by OCB. Thus, the study results show how SRHRM can promote employee engagement with their organisations for higher bank reputation and decrease turnover intention's adverse effects in the Bangladeshi banking sector. The call for SRHRM is becoming significant for banks and their stakeholders; however, little has been known about this practice in the context of south Asia and particularly in Bangladesh. SRHRM linked to CSR research has been performed mostly in corporate governance and strategy contexts and is a new addition to the literature of HRM [78–81]. This paper contributes as a catalyst for further empirical studies. Hence, it will enhance future knowledge to answer why and when SRHRM is likely to lead to positive outcomes for employees, organisations, and societies.

**Author Contributions:** Conceptualization, F.A.S., A.H. and S.R.; methodology, F.A.S., A.H. and S.R.; software, A.H.; validation, F.A.S., A.H. and S.R.; formal analysis, A.H.; investigation, F.A.S., A.H. and S.R.; resources, F.A.S., A.H. and S.R.; data curation, F.A.S., A.H. and S.R.; writing—original draft preparation, F.A.S., A.H. and S.R.; writing—review and editing, F.A.S., A.H. and S.R.; visualization, F.A.S., A.H. and S.R.; supervision, F.A.S., A.H. and S.R.; project administration, F.A.S., A.H. and S.R.; funding acquisition, F.A.S., A.H. and S.R. All authors have read and agreed to the published version of the manuscript.

**Funding:** This study was funded by the Institute for Advanced Research (IAR), United International University (UIU), Bangladesh.

**Institutional Review Board Statement:** Not applicable.

**Informed Consent Statement:** Not applicable.

**Data Availability Statement:** Not applicable.

**Conflicts of Interest:** The authors declare no conflict of interest.

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
