# Peer review of "Socially Responsible HRM, Employee Attitude, and Bank Reputation: The Rise of CSR in Bangladesh"

_sustainability, doi:10.3390/su13052753_

Round 1

Reviewer 1 Report

This paper provides with novel insights into the organizational culture and norms in banks. This is example of well-thought, detailed, and well-structured scientific writing, which should be appreciated. The outcomes are fresh, interesting, and multiple. The authors' awareness of the literature is very good. The paper requires only small amendments before acceptance.

  • Please, state your objective more clearly in Introduction.
  • Table 1, 2: please, indicate sources.
  • Sub-section 5.1: please, give people earnings in both BDT and USD.
  • Sub-section 5.2: please, give citations to 2-3 basic sources on the Likert-type scales.
  • The text passage between the title of the section 7 and the title of the sub-section 7.1 is lengthy, and it MUST be specified as a sub-section 7.1, with subsequent re-numbering of the following sub-sections of this section.
  • Conclusions should bear a numbered list of 3-7 main findings (from Results and Discussion) communicated in easy form.
  • Please, consider possibility to check and to cite these sources and some literature cited in them:

https://www.sciencedirect.com/science/article/pii/S2214462515000146

https://www.sciencedirect.com/science/article/pii/S1059056020301635

https://www.sciencedirect.com/science/article/pii/S1042444X20300013

https://www.sciencedirect.com/science/article/pii/S2314721017301068

https://www.mdpi.com/2071-1050/11/13/3515

https://www.mdpi.com/1911-8074/14/2/46/htm

  • The writing is clear, but it needs linguistic polishing.

Reviewer 2 Report

Thank you very much for the opportunity to review this paper, I have found it very interesting. The topic is relevant and of current significance. However, I would suggest the authors some revisions to improve the clarity of the paper.

Inconsistent listing of sources e.g. lines 108, 87,… But there are also shortcomings in the references, sources 16, 17, 18, 37.

Some abbreviations seem unnecessary, e.g. GOB. A large number of abbreviations make the paper difficult to read and reduce the comprehensibility of the paper.

The authors could better explain why they focused on employee’s perceived bank reputation in Hypothesis 3. For example, extend this part how it relates to Perceived Organizational Performance in Recruiting and Retaining Employees and Sustainable Human Resource Management.

The model shown in Figure 1 is confusing for me. I am not sure whether the research methods have made it possible to prove whether it is a direct or mediated relationship.

The paragraphs in subchapter 5.2 are too short to be separate numbered subchapters. I recommend removing the numbering.

Figure 2 needs to be modified. The data in the frame are incomplete.

Non-uniform abbreviations are also used (SRHRM and SR-HRM)

Some statements are unusual, e.g. line 439: Hypothesis 7 (H7) recommends ....?

I also recommend to edit the English language.

Reviewer 3 Report

Socially responsible human resource management in developing nations is an important area of inquiry.  This article adds incrementally to work done in this area.  This is a concise study with significant implications.  

That being said, I recommend adding to the discussion by suggesting what specifically managers could do to improve SRHRM practices.  

Reviewer 4 Report

This is an interesting article that investigates the association among socially responsible HRM, organizational citizenship behavior, turnover intention and bank reputation.

The research gap is identified by the authors; however, I believe it may be relevant if the authors clearly define a research question (in the introduction section) which allow to fill the gap in the literature.

I was able to verify with satisfaction that the investigation is empirically robust and that the authors use an adequate research design.

While the article is well structured and organized, the conclusions are the Achilles' heel of the article, as they are not balanced when compared with the other sections. Thus, I also think the article would benefit if the authors could divide the conclusion the following subsections: - contributions to management theory and practice; - research limitations; - recommendation for future research.

Round 2

Reviewer 4 Report

I think the article is better, clearer and makes a stronger contribution. My recommendation is to Approve the article.

Author Response

Many thanks!